# Pathways to Sustainable Development: Corporate Digital Transformation and Environmental Performance in China

**Pingguo Xu** [1] , **Leyi Chen** [1] **and Huajuan Dai** [2,*]

[1]  School of Economics and Trade, Hunan University, Changsha 410079, China
[2]  Business School, Hunan First Normal University, Changsha 410002, China
[*]  Correspondence: daihuajuan@hnfnu.edu.cn

**Abstract:** Environmental pollution remains a serious sustainable development issue. Enterprises, as important agents of sustainable development, are receiving increasing attention regarding their efforts to protect the environment. The rapid development of the digital economy has become a new driver of corporate environmental governance and environmental performance improvements, marking a new sustainable development path. We study the environmental effects of corporate digital transformation from the perspective of environmental performance using Chinese A-share listed companies. We construct a two-step systematic GMM econometric model and find that corporate digital transformation significantly improves environmental performance. Heterogeneity analysis shows that environmental performance improvement through digital transformation is more pronounced among state-owned, large, and heavily polluting enterprises. Mechanistic analysis shows that corporate digital transformation mainly improves environmental performance by enhancing green technological innovation and corporate governance. Further analysis shows a nonlinear relationship between corporate digital transformation and environmental performance. The research not only analyzes the impact of corporate digital transformation on environmental performance from multiple dimensions but also discovers the transmission mechanism of digital transformation that affects environmental performance and verifies a possible nonlinear relationship, providing a theoretical basis and practical reference for promoting corporate digital transformation and sustainable development.

**Keywords:** sustainable development; corporate digital transformation; environmental performance; green technology innovation; corporate governance

## 1. Introduction

In many countries around the world, it is still difficult to balance energy consumption, carbon emissions, and economic development. Economic development and its associated industrialization and urbanization often increase the pressure on the ecological environment, and this environmental pressure is particularly prominent in developing countries. According to the 2022 Global Environmental Performance Index (EPI), jointly released by Yale University and Columbia University, environmental performance varies significantly across countries globally, with South and Southeast Asian countries performing poorly and China ranking 160th in environmental performance. The EPI is a powerful policy tool that reflects how well governments are implementing environmental goals and can help countries identify problems so that they can develop better environmental policies to support efforts to achieve the UN Sustainable Development Goals and move society toward a sustainable future. Since its reform and opening up, China has experienced rapid economic development, with its economy leaping to the position of second largest in the world; however, the accompanying environmental problems have always been a major challenge. In September 2020, the Chinese government clearly proposed a "carbon peak" by 2030 and "carbon neutral" by 2060." These milestones mean that by 2030, carbon

dioxide emissions per unit of GDP will drop by more than 65% compared with 2005, and carbon dioxide emissions will reach their peak and be steadily reduced. By 2060, the green, low-carbon cycle development of the economic system and clean, low-carbon, safe and efficient energy system will be fully established, energy utilization efficiency will reach the advanced international level, the proportion of nonfossil energy consumption will be more than 80%, and the goal of carbon neutrality will be achieved.

With the rapid development of technological advances and network digitization, the digital economy—as a new driver of quality economic development with its high penetration, scale effect, and network effect—has represented an immediate response to the dramatic changes in the internal endowments of economies and the external environment in the new development landscape and is driven by the digital revolution and emerging technologies that increase internet investments, facilitate the construction of big data models and the application of artificial intelligence, consolidate information technology, and change how society produces and operates [1]. China's digital economy is developing rapidly in terms of total volume and structure; statistics show that China's annual software information technology services business revenue grew by 17.7% in 2021, industrial robots of enterprises larger than a designated size reached 30.8% year-on-year, 3D printing equipment grew by 27.7% year-on-year, the number of industrial internet platforms exceeds 150, more than 2000 "5G + industrial internet" projects were under construction, 20 typical application scenarios and 10 practical activities in key industry areas were formed, the level of innovation and application is in the first echelon globally, and 5G mobile communication technology, equipment and applications, big data, cloud computing, blockchain, and other technologies are in leading positions, a total of 1.425 million 5G base stations were made operational, and the number of 5G cell phone users reached 355 million. From 2012 to 2021, the size of China's digital economy grew from 11 trillion RMB to over 45 trillion RMB, and the proportion of the digital economy to GDP increased from 21.6% to 39.8%. This ambitious development of the digital economy provides a new engine and direction for environmental sustainability and is an important catalyst for achieving the dual carbon goals of carbon peaking and carbon neutrality. Existing studies on the impact of the digital economy on the environment are also becoming increasingly richer, and a large number of scholars study the environmental effects brought by the digital economy from different dimensions. For example, the information and knowledge sharing brought by the digital economy reduce operational costs and improve the efficiency of pollution management, and the development of digital technology promotes improvements in green innovation. The construction of digital platforms is also changing the traditional marketing method, which improves the efficiency of transactions over time. The construction of digital platforms has changed traditional marketing methods, improved the efficiency of transactions in time and space, and reduced pollution emitted through the marketing process. Information and Communication Technology (ICT) can be a tool for environmental sustainability [2]. Arguably, the digital economy not only improves the efficiency of environmental governance through information technology but also causes energy rebound effects and intensifies pollutant emissions through scale expansions. The digital economy and carbon emissions show a nonlinear relationship [3], and the digital economy does not always show a positive, linear effect on environmental sustainability [4,5]. Achieving environmental sustainability is a challenge that needs to be faced at present and even for a long time in the future, while the digital economy represents a new path for environmental sustainability in the new era, companies in particular need to enhance their digital capabilities and balance their economic, environmental and social impacts [6]. Based on the perspective of environmental performance, this paper studies the environmental effects of corporate digital transformation and explores the internal logical relationship between digital transformation and environmental performance, which has important theoretical and practical significance for achieving environmental sustainability.

The marginal contributions of this paper are mainly reflected in the following aspects. First, the impact of digital transformation on environmental performance is studied at the

corporate level. Textual analysis is used to measure corporate digital transformation and corporate environmental capital expenditures are scaled to environmental performance with more adequate and reasonable sample data that better reflects the contribution of enterprises in environmental governance and protection and provides a new entry point for the study of corporate digital transformation and environmental performance. Second, the environmental effects of corporate digital transformation are analyzed in multiple dimensions from the corporate and regional levels, providing more practical experience for corporate sustainability. Third, the transmission mechanism of the impact of digital transformation on environmental performance is explained from the microscopic perspective of enterprises themselves based on green technological innovation and corporate governance, enriching their inner logical connection. Fourth, the possible nonlinear impact between corporate digital transformation and the environment in existing studies is investigated, and the nonlinear relationship between the two is verified, which expands the scope of existing studies on corporate digital transformation and environmental performance.

The rest of the paper is organized as follows. The second part presents a review of the relevant literature. The third part presents the theoretical analysis and hypothesis of corporate digital transformation on environmental performance. The fourth part presents the research methodology, variable selection, and data sources. The fifth part presents the empirical results and analyses. The sixth part presents the research conclusions and policy recommendations.

## 2. Literature Review

### 2.1. Factors Influencing Environmental Performance

Environmental performance can be assessed through a range of indicators, such as the prudent use of resources, pollution prevention, and waste reduction [7–9]. Corporate environmental performance reflects the impact of business activities on the natural environment and demonstrates the extent to which firms are committed to eco-friendly actions to protect the natural environment [10,11]. The literature on the factors influencing environmental performance is mainly addressed at the firm and society levels. At the firm level, corporate environmental responsibility has a positive impact on environmental performance [12]; for example, corporate environmental responsibility promotes green innovation and environmental performance, which in turn strongly enhance environmental performance [13], and corporate social responsibility (CSR) practices and the financing of various environmental projects improve the environmental performance of organizations and ultimately contribute to sustainable development [14]. The level of corporate governance similarly plays an important role in environmental sustainability; in fact, corporate board independence and board gender diversity are positively associated with carbon reduction initiatives, with higher board independence and larger companies tending to have higher environmental performance [15]. Interestingly, if a board member's reputation is damaged, then environmental performance is also higher when the problematic director seeks to rebuild his or her reputation, and companies run by problem directors score higher on environmental management and environmental reputation than nonproblem director affiliates [16]. Arguably, corporate carbon disclosures tend to indicate their underlying actual carbon performance [17]. Aggressive corporate environmental strategies predict corporate environmental performance through green product innovation [18]; however, price competition faced by firms is thought to reduce environmental performance by shortening the time horizon of its application through strategic decisions [19], companies can also support managers and entrepreneurs in achieving improvements in corporate environmental performance by building Sustainable Service Innovation (SOSI) tools [20]. On a societal level, the relationship between green supply management and environmental performance becomes more evident when environmental and reputational advantages are undervalued [21], and green entrepreneurial orientation can improve environmental performance, which improves financial and environmental performance through the introduction of environmentally friendly products and services. However, international trade frictions

can significantly reduce environmental performance; for example, the implementation of unilateral and multilateral trade policies and economic sanctions can have a significant negative impact on the environmental performance of the host country [22]. Environmental taxation can similarly limit environmental performance [23].

*2.2. The Impact of Digital Economy on Enterprise Development*

Digital transformation has improved information sharing and integration, bringing about digital technology spillover effects; it is necessary for companies to do business in most markets, contributing to sustainable business development and industry benefits [24]. The digital economy promotes green technology innovation in both time and space, mainly through technical efficiency and technology gaps [25], and can improve such innovation by alleviating financing constraints and attracting government subsidies [26]. Digital finance, as an important component of the digital economy, can significantly improve the quantity and quality of green innovation by alleviating corporate financial constraints and giving full play to internal and external information effects [27]. Some studies have also concluded that the impact of the digital economy on green total factor energy efficiency significantly shifts from negative to positive with the development of the digital economy, and there is a nonlinear relationship between them [28]. In terms of CSR, arguably, digitalization has opened up new ways to facilitate the socialization process of all types of organizations, especially for corporate social development [29]. The development of digital finance significantly reduces the cost of debt financing, which increases CSR by reducing information asymmetry in financial markets, thus facilitating corporate access to financial markets and effectively alleviating their financing constraints, enabling them to invest more capital in CSR [30]. From the perspective of corporate governance, although digital intensity has a u-shaped correlation with profit-oriented financial performance [31], the impact of digital transformation on corporate internal auditing is considered more an opportunity than a constraint [32], where digitalization improves corporate auditing and corporate governance by promoting a culture of innovation within companies through audits [33]. Moreover, digital informatics and digital technologies, such as artificial intelligence, big data, and blockchain analytics, can effectively assist in developing better and more transparent relationships, weakening principal-agent conflicts, and forming more rational boards of directors—good corporate governance means that companies have real-time access to information on key activities to gain a competitive advantage through the use of shared digital databases, email, video conferencing, the internet of things, and workstation technologies [34,35].

*2.3. Impact of Corporate Digital Transformation on Environmental Performance*

Existing studies on the impact of the digital economy on environmental performance are mainly from the perspective of carbon emissions, and the vast majority of them agree that the digital economy suppresses carbon emissions in general [36]. However, these studies have different views on the study of intermediate transmission mechanisms; for example, some studies in the literature argue that the overall development of the digital economy can suppress regional carbon emissions through industrial progress and the optimization of energy consumption [37]. Others argue that the digital economy can suppress regional carbon emissions mainly through the innovation effect and industrial structure upgrading effect [38,39]. It has also been found that R&D investment not only suppresses emission levels but also plays a moderating role between digitalization and CO2 emissions [40]. Similarly, green energy efficiency is also considered to moderate the effect of the digital economy on carbon emissions, which promotes carbon emissions when green energy efficiency is low and reduces carbon emissions when green energy efficiency is high [41]. These studies are based on the linear relationship between the digital economy and carbon emissions; in fact, although the development of the digital economy can significantly reduce the intensity of carbon emissions, the per capita carbon emissions increase as a result [42]. If the digital economy reduces carbon emissions by increasing energy intensity, it increases carbon emissions by promoting economic expansion [43]. A

growing number of studies show that digital technologies can successfully improve the efficiency of carbon emissions but may not reduce total carbon emissions [44,45]. The impact of digital transformation on energy use is not a simple linear relationship; digital development increases carbon emissions by expanding the scale of production and the energy rebound effect, showing a "U" shaped nonlinear relationship, and when digital transformation reaches a certain level, resource efficiency gains occur [46,47].

In summary, the literature on environmental performance mainly focuses on the impact factors, while studies on corporate digital transformation mainly focus on the impact of digital economic development on carbon emissions and mostly stay at the city and provincial levels, with fewer studies at the enterprise level. The literature on the environmental effects of the digital economy from corporate environmental performance studies is even scarcer, and whether the digital economy always improves the environment remains controversial. Most studies support a positive linear relationship between the digital economy and the environment, and a few studies suggest that there may be a certain nonlinear relationship between the impact of the digital economy on the environment. Is the impact of corporate digital transformation on environmental performance linear or nonlinear? What are the transmission mechanisms? This paper attempts to answer these questions.

## 3. Theoretical Analysis and Hypotheses

Corporate digital transformation is a new driver of sustainability that contributes to environmental performance, and we think that corporate digital transformation affects environmental performance through two channels: green technology innovation and corporate governance.

### 3.1. Green Technology Innovation

Corporate digital transformation is a new type of drive to reorganize and optimize production resources with the help of technological innovation by interembedding digital technology and traditional production models, breaking through the boundary constraints of traditional elements, and showing a new value function [48]. Corporate digital transformation promotes corporate green innovation and environmental performance through information sharing, big data applications, enhancing resource acquisition capability, and optimizing and upgrading business models. From the perspective of information sharing, corporate digital transformation can promote the transmission and exchange of information related to internal and external resources and the environment by enhancing the level of information sharing by enterprises and can improve the transmission path of innovation capability and energy-environmental performance [49–52]. Through efficient information transfers and knowledge accumulation, corporate digital transformation can also stimulate more open innovation practices, enhance industry 4.0 technology, stimulate enterprise innovation [53], and improve labor productivity, all of which promote corporate green technology innovation [54], resulting in increased innovation output and efficiency, improved environmental performance, and motivation for enterprises to increase their engagement in green innovation activities [55,56]. From the perspective of big data applications, digital transformation can promote the wide application of big data in enterprises. The wide application of big data enables enterprises to gain greater technological innovation and technological advantages, which further enhances their competitive advantages, improves their corporate value, drives their carbon emission performance and capacity utilization, and reduces energy consumption, thus improving environmental performance [57]. From the perspective of resource acquisition capability, digital transformation also enhances the ability of enterprises to acquire resources required for operations, production, and innovation activities to reduce internal and external operating costs and energy utilization costs [58,59], improve energy utilization and green output [60], and realize the transformation of enterprises from asset-heavy to asset-light types, all of which improves environmental performance. From the perspective of optimizing and upgrading business

models, digital transformation also optimizes and upgrades traditional production tools and business models and promotes a specialized division of labor that contributes to the establishment of an open digital innovation system. In turn, this open digital innovation system transforms the enterprise innovation model from a traditional closed model to an open innovation model with broad participation from various departments and even the entire industrial chain and consumers. Thus, integrated and networked innovation is realized, the specialized production division of labor within enterprises is accelerated, green technology innovation is further promoted and environmental performance is improved. Accordingly, this paper proposes the following hypothesis.

**Hypothesis 1.** *Corporate digital transformation improves environmental performance by promoting green technology innovation.*

*3.2. Corporate Governance*

Corporate governance includes business management, organizational structure optimization, and other activities that are the intrinsic drivers of a firm's operational development. The quality of corporate governance directly affects the future sustainability of the enterprise and is a lever to promote sustainable development [61]. However, the internal conflicts resulting from the separation of corporate control and ownership are direct causes of corporate management and operational efficiency. Executives tend to neglect long-term business and environmental performance resource management in pursuit of short-term business performance and personal wealth accumulation, which makes enterprises expand production in the short term, overuse resources, and overdraw on their future. In contrast, a reasonably sized board of directors can significantly improve the efficiency of corporate governance. Corporate digital transformation promotes corporate governance, which is conducive to long-term corporate sustainability. In terms of corporate organizational structure, corporate digital transformation promotes the optimization and upgrading of the organizational structure and forms a good organizational structure for the board of directors, enabling enterprises to plan their sustainable development path from a reasonable degree of carbon information disclosure, increase the use of renewable resources, and obtain higher environmental performance [62]. Good corporate governance can also manage the legitimacy of regulatory policy changes, increase environmental information disclosures, improve environmental performance, alleviate corporate environmental pressure [63], and to some extent resolve divergent interests among stakeholders and managers regarding environmental activities [64]. From the perspective of CSR, digital finance development significantly reduces the cost of debt financing and promotes CSR. As increased environmental regulations have put forward higher requirements for corporate environmental performance, enterprises in the environmental pollution industry face strict environmental assessments by capital markets and for loans from financial institutions, and their financing constraints are greater for poor environmental performance, preventing them from expanding production and value creation. The development of digital finance has enhanced the ability of enterprises to obtain information and knowledge, reduced financing constraints in time and space, made it easier to obtain financing, effectively weakened the principal-agent conflict, solved the problem of less financing or difficult financing, improved CSR, rationalized and enabled transparent corporate governance, and improved environmental performance. The refinement and intelligence of digital technology have strengthened enterprise supervision and management, especially in the case of weak internal controls and low institutional ownership. Digital transformations strengthen corporate governance supervision, increase information transparency, complement corporate governance mechanisms, and improve enterprise environmental performance [65]. Based on this, this paper proposes the following hypothesis:

**Hypothesis 2.** *The digital transformation of enterprises improves environmental performance by enhancing corporate governance.*

Overall, corporate digital transformation mainly affects environmental performance through two channels: improving green technology innovation and corporate governance. The corresponding logical framework is shown in Figure 1.

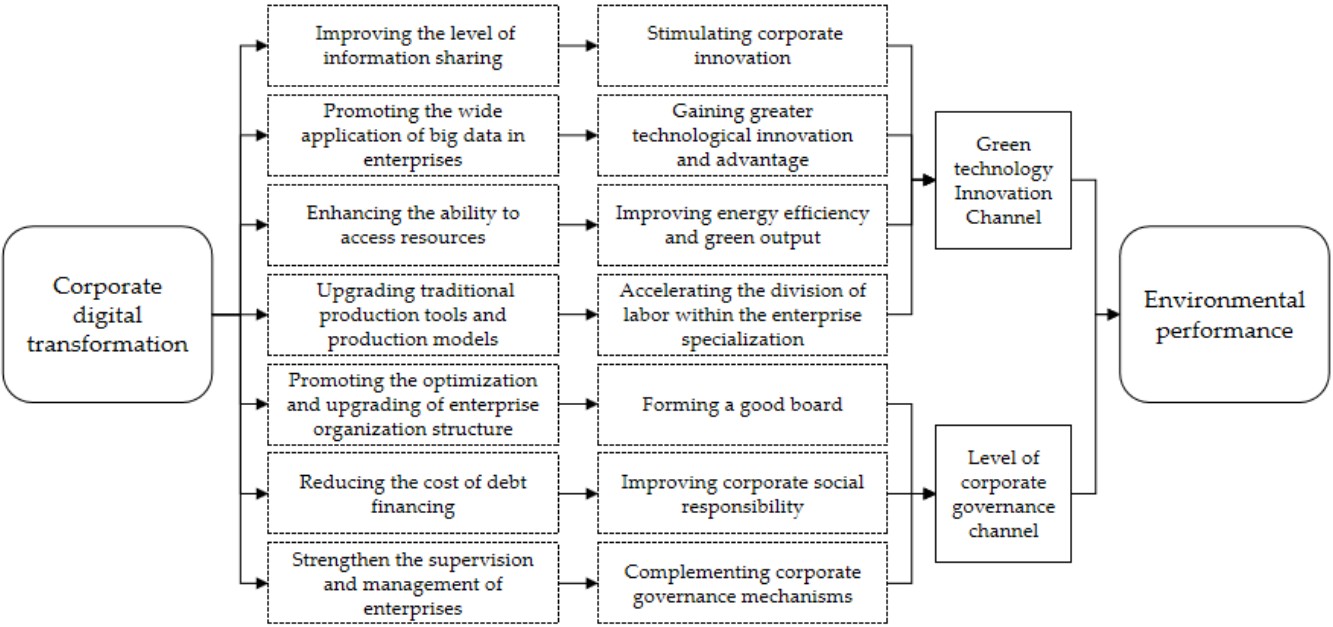

**Figure 1.** Logical Framework for Enterprise Digital Transformation Impacting Environmental Performance.

## 4. Research Methodology, Variable Selection, and Data Sources

### 4.1. Empirical Model Setting

In consideration of the possible mutual causal relationship between digital transformation and environmental performance and to avoid endogeneity problems caused by omitted variables, this paper uses a two-step systematic GMM econometric estimation method for benchmark regression and constructs the following regression model.

$$Enp_{i,t} = \alpha_0 + \alpha_1 Enp_{i,t-1} + \alpha_2 Dig_{i,t} + \alpha_3 X_{i,t} + \varepsilon_{i,t} \tag{1}$$

$Enp_{i,t}$ denotes the environmental performance of enterprise $i$ in year $t$, and the first-order lag of environmental performance $Enp_{i,t-1}$ is used as an explanatory variable in the model to eliminate the possible correlation between the explanatory variables and the random disturbance term when the lag term is not introduced. $Dig_{i,t}$ denotes the digital transformation level of enterprise $i$ in year $t$, $X_{i,t}$ denotes the indicators of enterprise $i$ in year $t$, which are the control variables of this paper, and $\varepsilon_{i,t}$ are random disturbance terms.

### 4.2. Description of Variables

#### 4.2.1. Explained Variables

Environmental performance (Enp). Corporate environmental performance reflects the impact of business activities on the natural environment and demonstrates the extent to which enterprises are committed to eco-friendly actions to protect the natural environment [10]. This information can be obtained by assessing and scoring the good or bad pollution emissions associated with corporate environmental performance [66]. Due to the fact that the subjective arbitrariness of such measured environmental performance assignments, studies have used actual environmental capital expenditures of enterprises to directly indicate corporate environmental performance [67], or the proportion of corporate emissions expenditures in total operating income. Compared to a single environmental performance indicator, corporate capital expenditures for environmental protection are a more accurate indicator [68]. Therefore, in this paper, the proportion of corporate emissions

expenditures in total operating income is used to measure corporate environmental performance. A larger value of the indicator represents more environmental pollution emissions and lower environmental performance, and vice versa.

### 4.2.2. Explanatory Variable

Corporate digital transformation (Dig). The literature on the measurement of corporate digital transformation is mainly based on comprehensive indicators of various components of the digital economy, such as digital economy foundation, digital industrialization, industrial digitization, and digital economy penetration [36–38], involving the internet penetration rate, number of cell phone base stations, total telecommunication services, fixed asset investments in information transmission and computer services, and software business income. Since there are fewer disclosures about digital economy-related information by Chinese listed companies, some scholars have also used textual analysis methods to measure the digital development of enterprises by counting the frequency of words related to digitalization in the annual reports of listed companies [69,70]. In this paper, we also measure the degree of digitalization of microenterprises by counting the frequency of digitalization-related words in the annual reports of Chinese listed companies and dividing the sum of this count by the length of the text in the "Management's Discussion and Analysis" (MD and A) section of the company's annual report. A higher indicator value represents a higher degree of corporate digital transformation. To ensure the robustness of the study results, this paper also conducted a principal component analysis on digital transformation-related internet broadband access ports, cell phone penetration rate, internet broadband access users, mobile internet users, year-end resident population, internet access port density, and mobile internet penetration rate to obtain comprehensive indicators of enterprise digital transformation [37,38], which were tested for robustness.

### 4.2.3. Mechanism Variables

Green technology innovation (GRT). Green technological innovation is usually measured by the ratio of R&D investment to energy consumption or pollutant emissions [71]; however, green patents reflect green innovation activities more objectively and accurately [72]. Therefore, in this paper, we choose the number of green patent applications to characterize corporate green innovation and adopt the number of corporate green invention patent applications to express green technological innovation [72–74]. The robustness test is carried out with the number of green utility model patent applications. To remove the time trend and facilitate the latter analysis, we add 1 to the patent data, take the natural logarithm, and divide it by 100.

Corporate governance (CGO). Corporate governance comprises the duties and responsibilities that a company's board of directors must carry out to successfully lead the company and maintain relationships with shareholders and other stakeholders. CEO power, board capital, and ownership structure are important components of corporate governance [62], and board profile is also important from an agency theory perspective, as it enables the institution to engage in opportunistic activities due to its dominant position. Corporate governance also includes information disclosure, financial transparency, equity balance, separation of two rights, and management shareholding ratio. This paper uses equity balance, separation of two rights, and management shareholding ratio to measure corporate governance.

### 4.2.4. Control Variables

This paper further controls the following firm variables: firm size (Size), measured as the natural logarithm of annual total assets divided by 100; net profit margin on total assets (Roa), measured by dividing net profit by the average balance of total assets; return on net assets (Roe), measured by dividing net income by the average balance of shareholders' equity; total asset turnover ratio (Ato), measured by dividing operating income by average total assets; cash flow ratio (Cash), measured by dividing net cash flow from operating

activities by operating income and then dividing by 100; operating income growth rate (Grow), measured by dividing the current year's operating income by the previous year's operating income minus 1; and percentage of independent directors (Indep), measured by dividing the number of independent directors by the number of directors and then dividing by 100. The variables are defined as shown in Table 1.

**Table 1.** Variable Definition.

| | Variable | Definition |
|---|---|---|
| Explained variable | Enp | Enterprise's expenditure on sewage charges/total operating revenue |
| Explanatory variable | Dig | Total number of digitally relevant terms/text length of the MD and A section of the company's annual report |
| | GRT | Ln(number of green invention patent applications by enterprises + 1)/100 |
| Mechanism variables | CGO | Balance: Sum of shareholdings of the second to fifth largest shareholders divided by shareholding of the first largest shareholder |
| | | Separation: Difference between the ratio of control and ownership |
| | | Mshare: Management shareholding data/total share capital |
| | Size | Ln(total assets)/100 |
| | Roa | Net income/average balance of total assets |
| | Roe | Net income/average balance of shareholders' equity |
| Control variables | Ato | Operating income/average total assets |
| | Cash | (Net cash flow from operating activities/operating income)/100 |
| | Grow | (Current year's operating income/Prior year's operating income)—1 |
| | Indep | (Number of independent directors/number of directors)/100 |

### 4.3. Data Source

In this paper, the data on 1268 A-shares listed companies on the Shanghai and Shenzhen exchanges from 2010 to 2019 were selected as the initial research sample and were processed as follows. First, financial companies were excluded; second, the ST and period delisting samples were excluded; and third, companies that conducted an IPO in the years under examination were excluded. The descriptive statistics of the variables are shown in Table 2.

**Table 2.** Descriptive Statistics.

| Variable | Obs | Mean | Std. Dev. | Min | Max |
|---|---|---|---|---|---|
| Enp | 2124 | 0.0019 | 0.0038 | 0.0000 | 0.0771 |
| Dig | 2124 | 0.0001 | 0.0005 | 0.0000 | 0.0071 |
| Roa | 2124 | 0.0315 | 0.0611 | −0.2466 | 0.2286 |
| Roe | 2124 | 0.0513 | 0.1417 | −0.6993 | 0.3829 |
| Ato | 2124 | 0.7094 | 0.4323 | 0.0656 | 2.6489 |
| Size | 2124 | 0.2250 | 0.0128 | 0.1930 | 0.2594 |
| Indep | 2124 | 0.3686 | 0.0516 | 0.2500 | 0.5714 |
| Cash | 2124 | 0.0009 | 0.0015 | −0.0074 | 0.0071 |

## 5. Empirical Results and Analyses

### 5.1. Results of the Benchmark Model

Table 3 reports the results of the benchmark regressions of corporate digital transformation on environmental performance. Columns (1) and (2) report the regression results of the systematic GMM without and with the addition of control variables, and columns (3) and (4) report the regression results of the differential GMM without and with the addition of control variables, respectively. The regression results show that the regression coefficient of corporate digital transformation is significantly negative at the 1% significance level, indicating that corporate digital transformation can effectively reduce corporate pollution emissions and improve corporate environmental performance. The regression coefficient of the lagged first-order of environmental performance on environmental performance is

significantly positive at the 1% significance level, indicating that environmental pollution emissions in the previous year positively affect the current period; in other words, the improvement in environmental performance in the previous year is beneficial to the improvement in environmental performance in the current period. As a consistent estimation, GMM is established on the premise that there is no autocorrelation of second and higher orders in the residual series in the difference equation, and the instrumental variables are strictly exogenous; thus, the estimation results need to be subjected to the Arellano-Bond serial correlation test and the Sargan test. From the regression results of each column, the $p$-value of the Sargan test is greater than 0.1, indicating that the new tool variables of the two-step system GMM and the differential GMM are effective, and there is no overidentification problem. In the residual series correlation test, the $p$- value of AR (2) is greater than 0.1 and of AR (1) is less than 0.1, indicating that there is a second-order autocorrelation in the residuals after differencing and no first-order autocorrelation.

**Table 3.** Benchmark Regression.

| Variables | Enp | | | |
|---|---|---|---|---|
| | **(1)** | **(2)** | **(3)** | **(4)** |
| | **SYS-GMM** | | **DIFF-GMM** | |
| $Enp_{t-1}$ | 0.6389 *** | 0.6372 *** | 0.4645 *** | 0.3131 *** |
| | (0.0192) | (0.0298) | (0.0160) | (0.0413) |
| Dig | −0.2889 *** | −0.3574 *** | −0.3234 *** | −0.2132 *** |
| | (0.0155) | (0.0323) | (0.0208) | (0.0462) |
| Roa | | 0.0004 | | −0.0014 |
| | | (0.0013) | | (0.0013) |
| Roe | | −0.0009 * | | −0.0004 |
| | | (0.0005) | | (0.0004) |
| Ato | | −0.0008 *** | | −0.0010 *** |
| | | (0.0001) | | (0.0001) |
| Size | | −0.0240 *** | | −0.0498 *** |
| | | (0.0050) | | (0.0106) |
| Indep | | 0.0027 *** | | 0.0012 ** |
| | | (0.0004) | | (0.0005) |
| Cash | | −0.0037 | | −0.0414 *** |
| | | (0.0086) | | (0.0092) |
| _cons | 0.0005 *** | 0.0056 *** | 0.0008 *** | 0.0127 *** |
| | (0.0000) | (0.0012) | (0.0001) | (0.0024) |
| Controls | YES | YES | YES | YES |
| AR(1) | 0.0016 | 0.0016 | 0.0061 | 0.0378 |
| AR(2) | 0.2425 | 0.2819 | 0.2204 | 0.3083 |
| Sargan-test | 0.1451 | 0.3507 | 0.1589 | 0.2492 |
| N | 1673 | 1673 | 1307 | 1307 |

Notes: Standard errors in parentheses, * $p < 0.1$, ** $p < 0.05$, *** $p < 0.01$, controls correspond to control variables, AR (1), and AR (2), and the Sargan test corresponds to the $p$-value.

*5.2. Robustness Checks*

To ensure the robustness of the empirical results in this paper, further robustness tests are conducted by replacing the explanatory and explained variables, adding significant control variables, and replacing the sample interval and measurement method.

Replacement of explanatory variables and explained variables. In column (1), the explanatory variable corporate digital transformation (Dig) is replaced, and the comprehensive indicators of corporate digital transformation are regained by conducting principal component analysis on digital transformation-related internet broadband access ports, cell phone penetration, internet broadband access subscribers, mobile internet subscribers, year-end resident population, internet access port density and mobile internet penetration. Column (2) shows the regression result of replacing the explained variable (Enp), and the ratio of total enterprise environmental protection expenditure to total

operating revenue is used to represent environmental performance, with larger values representing a higher percentage of environmental protection expenditures and worse environmental performance.

Adding control variables. Column (4) adds important control variables by adding the corporate gearing ratio (Lev) and Tobin's Q to the regression model. The gearing ratio (Lev) is a comprehensive indicator for evaluating the level of corporate indebtedness and measuring the ability of a company to utilize creditors' funds for operating activities and is measured by dividing total liabilities by total assets at the end of the year. Tobin's Q is an important indicator that measures the market value of enterprises. In this paper, we measure (value of circulating stock market + number of noncirculating shares x sum of net assets per share + book value of liabilities)/total capital of the enterprise.

Replacement sample interval. Column (4) replaces the data sample interval with 2011–2018.

Replacement of measurement method. To verify the robustness of the empirical estimation method, this paper also employs differential GMM for the estimation, and the regression results are presented in columns (3) and (4) of Table 3.

Table 4 reports the regression results of the robustness test. From the individual regression results, the regression coefficient of corporate digital transformation is still significantly negative, indicating that corporate digital transformation can effectively reduce environmental pollution emissions and improve the level of environmental performance. The Sargan test indicates that the model does not have an overidentification problem, and there is a second-order autocorrelation and no first-order autocorrelation in the residuals after differencing, which is consistent with the above empirical studies, indicating that the experimental findings of this paper are robust.

**Table 4.** Robustness Tests.

| Variables | Enp | | | |
|---|---|---|---|---|
| | (1) | (2) | (3) | (4) |
| $Enp_{t-1}$ | 0.7793 *** | 0.7650 *** | 0.6669 *** | 0.7119 *** |
| | (0.0414) | (0.0097) | (0.0220) | (0.0450) |
| Dig | −0.0990 ** | −0.2935 *** | −0.2798 *** | −0.2736 *** |
| | (0.0466) | (0.0551) | (0.0352) | (0.0686) |
| _cons | −0.0052 *** | 0.0096 *** | 0.0007 | 0.0005 |
| | (0.0017) | (0.0018) | (0.0010) | (0.0020) |
| Controls | YES | YES | YES | YES |
| AR (1) | 0.0649 | 0.0066 | 0.0015 | 0.0067 |
| AR (2) | 0.9929 | 0.9268 | 0.3456 | 0.6759 |
| Sargan-test | 1.0000 | 0.2902 | 0.2813 | 0.3138 |
| N | 207 | 1658 | 1673 | 1361 |

Notes: Standard errors in parentheses, ** $p < 0.05$, *** $p < 0.01$, Controls correspond to control variables, AR (1), AR (2), and the Sargan test corresponds to the $p$ value.

### 5.3. Heterogeneity Analysis

This paper further analyzes enterprise heterogeneity based on the nature of ownership (CNO), nature of size (STY), heavy pollution attributes (IFP), and regional heterogeneity. The regression results are shown in Tables 5 and 6.

**Table 5.** Heterogeneity Analysis of Corporate.

| Variables | (1) | (2) | (3) | (4) | (5) | (6) |
|---|---|---|---|---|---|---|
| | CNO | | STY | | IFP | |
| | POEs | SOEs | SMS | BIG | NHPs | HPs |
| $Enp_{t-1}$ | 0.5138 *** | 0.7880 *** | 0.6104 *** | 0.7104 *** | 0.7349 *** | 0.8321 *** |
| | (0.0165) | (0.0202) | (0.0041) | (0.0272) | (0.0042) | (0.0103) |
| Dig | −0.2165 *** | −0.3252 *** | 0.0078 | −0.2384 *** | −0.3523 *** | −0.4648 *** |
| | (0.0511) | (0.0494) | (0.0584) | (0.0337) | (0.0091) | (0.0860) |
| _cons | 0.0020 * | 0.0072 *** | 0.0066 *** | 0.0119 *** | −0.0007 *** | −0.0005 ** |
| | (0.0011) | (0.0011) | (0.0009) | (0.0012) | (0.0002) | (0.0002) |
| Controls | YES | YES | YES | YES | YES | YES |
| AR (1) | 0.0379 | 0.0016 | 0.1495 | 0.0032 | 0.0783 | 0.0039 |
| AR (2) | 0.5466 | 0.1650 | 0.2705 | 0.3884 | 0.6252 | 0.3084 |
| Sargan-test | 0.3104 | 0.8326 | 0.9986 | 0.3141 | 0.3388 | 0.6356 |
| N | 646 | 993 | 173 | 1497 | 476 | 1003 |

Notes: Standard errors in parentheses, * $p < 0.1$, ** $p < 0.05$, *** $p < 0.01$, controls correspond to control variables, AR (1), AR (2), and the Sargan test corresponds to the *p* value.

**Table 6.** Regional Heterogeneity Analysis.

| Variables | (1) | (2) | (3) |
|---|---|---|---|
| | East | Middle | West |
| $Enp_{t-1}$ | 0.6569 *** | 0.7881 *** | 0.8943 *** |
| | (0.0119) | (0.0037) | (0.0099) |
| Dig | −0.4431 *** | −0.1653 *** | −0.1828 *** |
| | (0.0201) | (0.0223) | (0.0153) |
| _cons | 0.0084 *** | 0.0040 *** | 0.0247 *** |
| | (0.0008) | (0.0005) | (0.0008) |
| Controls | YES | YES | YES |
| AR (1) | 0.0059 | 0.0142 | 0.0172 |
| AR (2) | 0.8358 | 0.5865 | 0.1265 |
| Sargan-test | 0.5516 | 0.7684 | 0.7594 |
| N | 890 | 431 | 352 |

Notes: Standard errors in parentheses, *** $p < 0.01$, controls correspond to control variables, AR (1), AR (2), and the Sargan test corresponds to the *p* value.

5.3.1. Enterprise Heterogeneity

Nature of enterprise ownership (CNO): Columns (1) and (2) of Table 5 show the regression results under the heterogeneous ownership nature of enterprises, with column (1) for private enterprises (POEs) and column (2) for state-owned enterprises (SOEs). From the regression results, the regression coefficient of corporate digital transformation remains significantly negative at the 1% significance level for both POEs and SOEs, and corporate digital transformation is beneficial for reducing environmental pollution emissions and improving environmental performance. The absolute value of the regression coefficient for SOEs is 0.3252, which is larger than the absolute value of the regression coefficient for POEs of 0.2165, indicating that the digital transformation of SOEs brings a greater increase in corporate environmental performance than do POEs. A possible reason is that to improve environmental performance, enterprises need more financing to obtain resources and technological innovation in the digital transformation process, and the government, to prioritize encouraging SOEs to improve their environmental performance, may use its policies at hand to compensate SOEs for better environmental performance [68]. SOEs receive more policy and incentive support from the government, and better promoting digital transformation brings greater environmental performance. In addition, SOEs are less capital intensive and have lower capacity utilization than POEs. While improving the

digital transformation of one unit, SOEs are able to reduce energy consumption to a greater extent, reduce carbon emissions, and improve corporate environmental performance.

Nature of firm size (STY): Columns (3) and (4) of Table 5 show the regression results under heterogeneous enterprise size. Column (3) reflects small and medium-sized enterprises (SMS), and column (4) reflects large enterprises (BIG). From the regression results, only the regression coefficient of the digital transformation of large enterprises is significantly negative at the 1% significance level. The regression coefficient of the digital transformation of SMS is positive and insignificant, indicating that digital transformation of large enterprises is beneficial to reducing environmental pollution and improving environmental performance, while digital transformation of SMS has an insignificant impact on environmental performance, which may be because large enterprises require more resources for production and operation consumption and production capacity scale. Digital transformation brings innovative technology development and management efficiency improvements, significantly improves the energy utilization rate, lowers energy consumption, significantly increases production capacity, more obviously reduces environmental pollution emissions, and improves environmental performance. In contrast, the production scale of SMS is smaller, the cost of resources and the financing needs of digital transformation may exceed its spillover effect in the short term, and the resulting environmental performance is poor.

Corporate Heavy pollution attributes (IFP): Columns (5) and (6) of Table 5 show the regression results of whether the enterprise is a heavy pollution type enterprise. Column (5) represents nonheavy pollution type enterprises (NHPs), and column (6) represents heavy pollution type enterprises (HPs). From the regression results, the regression coefficient of corporate digital transformation is still significantly negative at the 1% significance level regardless of whether or not enterprises are heavy polluting enterprises. Moreover, digital transformation is conducive to reducing environmental pollution and improving environmental performance, and the absolute value of the coefficient of the digital transformation of heavy polluting enterprises is 0.4648, which is larger than that of the coefficient of the digital transformation of nonheavy polluting enterprises of 0.3523. This result indicates that the digital transformation of heavily polluting enterprises can reduce environmental pollution emissions to a greater extent than that of other enterprises and can result in higher environmental performance, probably because heavily polluting enterprises are largely the target of key monitoring by environmental protection departments, their financing constraints and resource acquisitions are more difficult compared to those of nonheavily polluting enterprises, the relationship between sustainable development initiatives and corporate carbon performance is stronger [75], the need for better environmental performance is greater [76,77], the ability of enterprises to receive environmental assessment enhancements from environmental protection departments is related to their long-term future development, heavy polluting enterprises improve their environmental performance through the digital transformation of specialized and classified production operations, improved management efficiency, reduced energy consumption in more polluting segments, and improved green innovation technologies, while increasing the use of green environmental resources and reducing the scale from pollution emissions. In contrast, for nonheavily polluting enterprises, digital transformation has brought about improved environmental performance, but to a lesser extent than for heavily polluting enterprises.

### 5.3.2. Regional Heterogeneity

Using the geographic location of enterprises, this paper distinguishes their locations and analyzes the regional heterogeneity according to eastern, middle and western regions, respectively. The regression results are shown in Table 6. Columns (1), (2) and (3) of Table 6 correspond to the regression results for the eastern, middle and western regions, respectively. From the regression results, corporate digital transformation in the eastern, central and western regions can effectively suppress environmental pollution emissions and improve environmental performance, and it is significant at the 1% level. The inhibitory effect of corporate digital transformation on environmental pollution in the eastern region

is greater than that in the middle and western regions, while it is slightly greater in the western region than in the middle region. One reason may be that because the pollution emission and industrial energy consumption links were effectively controlled in the eastern region after industrial structure upgrades and industrial transfers to the middle and western regions, its digital technology level is higher than that of the middle and western regions, and the environmental effect of digital transformation is higher. As the degree of the industrial transfer of enterprises undertaking eastern pollution in the west is not as high as that in the middle region, the difficulty of environmental pollution control in the middle region is far greater than that in the east and west. The environmental effect of corporate digital transformation is less improved, and the degree of environmental performance improvements is lower than that in the east and west.

*5.4. Mechanism Analysis*

According to the previous theoretical analysis and hypothesis, corporate digital transformation mainly affects environmental performance through green technology innovation and corporate governance. In this paper, the impact of corporate digital transformation on environmental performance is further investigated through two channels: green technology innovation (GRT) and corporate governance level (CGO). The following econometric model is constructed for mechanism testing.

$$GRT_{i,t} = \beta_0 + \beta_1 GRT_{i,t-1} + \beta_2 Dig_{i,t} + \beta_3 X_{i,t} + \varepsilon_{i,t} \quad (2)$$

$$CGO_{i,t} = \gamma_0 + \gamma_1 CGO_{i,t-1} + \gamma_2 Dig_{i,t} + \gamma_3 X_{i,t} + \varepsilon_{i,t} \quad (3)$$

$GRT_{i,t}$ and $CGO_{i,t}$ denote the green technology innovation and corporate governance levels of enterprise $i$ in year $t$, and the first-order lagged sums of green technology innovation $CRT_{i,t-1}$ and corporate governance levels $CGO_{i,t-1}$ are placed in the model as explanatory variables to eliminate the possible correlation between the explanatory variables and the random disturbance terms when the lagged terms are not introduced. $Dig_{i,t}$ denotes the digital transformation level of enterprise $i$ in year $t$, and $X_{i,t}$ denotes the indicators of enterprise $i$ in year $t$, which are the control variables of the enterprise, and $\varepsilon_{i,t}$ are random disturbance terms.

5.4.1. Green Technology Innovation Channel (GRT)

Table 7 reports the regression results of the impact of corporate digital transformation on environmental performance through the green technology innovation channel. GRT1 is the number of corporate green utility model patent applications, and GRT2 is the number of corporate green invention patent applications. From the regression results in Table 7, the regression coefficient of corporate digital transformation is significantly positive at the 1% significance level, which indicates that corporate digital transformation promotes improvements in the green technology innovation level. According to the previous theoretical analysis, it is known that digital transformation improves labor productivity and promotes green technology innovation, and green innovation brings a fuller use of renewable energy and reduces energy consumption, which not only reduces environmental pollution but also improves energy utilization efficiency, reduces the cost of enterprises' energy utilization, and realizes the transformation of enterprises from asset-heavy to asset-light types, thus improving environmental performance. Improvements in green technology innovation can also promote the specialization of the enterprise's production division of labor, reduce pollution emissions in sectors with greater energy consumption, increase the release of capacity in sectors with better environmental benefits, promote the expansion of the scale of the green output of enterprises, and improve the environmental performance of enterprises. Hypothesis 1 is verified.

**Table 7.** Mechanism Analysis—Green Technology Innovation Channels.

| Variables | GRT1 | | GRT2 | |
|---|---|---|---|---|
| | **(1)** | **(2)** | **(3)** | **(4)** |
| $GRT1_{t-1}$ | 0.2653 *** | 0.2299 *** | | |
| | (0.0075) | (0.0013) | | |
| $GRT2_{t-1}$ | | | 0.3900 *** | 0.3207 *** |
| | | | (0.0199) | (0.0021) |
| Dig | 0.8260 *** | 0.6605 *** | 0.2491 *** | 0.6975 *** |
| | (0.0383) | (0.0099) | (0.0226) | (0.0142) |
| _cons | 0.0013 *** | −0.0314 *** | 0.0023 *** | −0.0343 *** |
| | (0.0001) | (0.0005) | (0.0002) | (0.0004) |
| Controls | NO | YES | NO | YES |
| AR (1) | 0.0000 | 0.0000 | 0.0000 | 0.0000 |
| AR (2) | 0.8039 | 0.4415 | 0.1131 | 0.2907 |
| Sargan-test | 0.1851 | 0.6718 | 0.4723 | 0.5864 |
| N | 1702 | 1702 | 1702 | 1702 |

Notes: Standard errors in parentheses, *** $p < 0.01$, controls correspond to control variables, AR (1), AR (2), and the Sargan test corresponds to the *p* value.

### 5.4.2. Corporate Governance (CGO) Channel

Equity balance, separation of powers and management shareholding ratio can well reflect the position of corporate control and ownership in corporate governance. In this paper, the three aspects of equity balance (Balan), separation of powers (Seper) and management shareholding ratio (Mshar) are used to indicate the level of corporate governance, which correspond to columns (1), (2) and (3) in Table 8, respectively. From the regression results, the regression coefficients of corporate digital transformation are significantly positive at the 1%, 5% and 1% significance levels, indicating that digital transformation can improve corporate governance. In fact, digital transformation reshapes new paths and mechanisms of corporate governance with high efficiency and innovative management, especially principal–agent costs and information asymmetry brought by the separation of powers (control and ownership). Digital transformation improves corporate productivity and performance through efficient investments and management efficiency and achieves a balance between the internal conflicts of control and the ownership of corporate managers. In addition, digital transformation makes data and information key factor inputs for enterprise production, and the data and information elements substantially improve the accuracy and effectiveness of factor resource allocation decisions throughout the enterprise and play a significant role in promoting the supervision of management to regulate their behavior and subjective judgments during the decision-making process or even in distorting the space for manipulation [78,79]. With improvements in big data capability, the gap between the innovation performance of enterprises in loose and tight modes increase significantly [80] to improve the level of corporate governance. The improvement in corporate governance has a multiplier effect on green technology innovation and the efficient production of enterprises. Enterprise management is more transparent and professional, management efficiency is further improved, sustainable green development is achieved, and the enterprise's environmental performance is improved, while enterprises with poor management generate fewer green patents, and ineffective corporate governance may constitute a major obstacle to environmental efficiency [81]. Hypothesis 2 is verified.

**Table 8.** Mechanism Analysis—Level of Corporate Governance.

| Variables | Balan | Seper | Mshar |
|---|---|---|---|
| | (1) | (2) | (3) |
| Balan$_{t-1}$ | 0.8416 *** | | |
| | (0.0110) | | |
| Seper$_{t-1}$ | | 0.6398 *** | |
| | | (0.0115) | |
| Mshar$_{t-1}$ | | | 0.8859 *** |
| | | | (0.0044) |
| Dig | 3.5559 *** | 2.9686 ** | 3.6994 *** |
| | (0.2614) | (1.4175) | (0.1856) |
| _cons | −0.0886 *** | −0.8874 *** | 0.0454 ** |
| | (0.0235) | (0.1160) | (0.0210) |
| Controls | YES | YES | YES |
| AR (1) | 0.0000 | 0.0000 | 0.0007 |
| AR (2) | 0.2661 | 0.2927 | 0.2591 |
| Sargan-test | 0.7405 | 0.3406 | 0.3088 |
| N | 1672 | 1612 | 1580 |

Notes: Standard errors in parentheses, ** $p < 0.05$, *** $p < 0.01$, controls correspond to control variables, AR (1), AR (2), and the Sargan test corresponds to the $p$ value.

### 5.5. Further Analysis

According to the previous section, digital economy development may increase carbon emissions by scaling up production and through energy rebound effects, and digital technologies can successfully increase the efficiency of carbon emissions [37] but may not reduce them [38]. The impact of digital transformation on energy use is not a simple linear relationship. To verify whether there is a nonlinear relationship between digital transformation and environmental performance, the following econometric model is constructed.

$$Enp_{i,t} = \delta_0 + \delta_1 Enp_{i,t-1} + \delta_2 Dig_{i,t} + \delta_3 Dig_{i,t}^2 + \delta_4 X_{i,t} + \varepsilon_{i,t} \tag{4}$$

$Enp_{i,t}$ denotes the environmental performance of enterprise $i$ in year $t$, and the first-order lag of environmental performance $Enp_{i,t-1}$ is placed as an explanatory variable in the model to eliminate the possible correlation between the explanatory variables and the random disturbance term when the lag term is not introduced. $Dig_{i,t}$ denotes the digital transformation level of enterprise $i$ in year $t$, and $Dig_{i,t}^2$ is the squared term of the firm's digital transformation indicators. $X_{i,t}$ denotes the indicators of enterprise $i$ in year $t$, which is the control variable of this paper, and $\varepsilon_{i,t}$ is the random disturbance term.

For the convenience of the analysis, the digital economy indicator Dig and the squared term indicator Dig2 are multiplied by 100 simultaneously, and the regression results are shown in Table 9. The primary term regression coefficient of corporate digital transformation is significantly negative at the 1% significance level, and the secondary term regression coefficient is significantly positive at the 1% significance level, indicating that there is a nonlinear relationship between the impact of corporate digital transformation and environmental performance. Digital transformation effectively reduces enterprise environmental pollution emissions and improves environmental performance before the inflection point; when it exceeds the inflection point, digital transformation instead intensifies corporate environmental pollution emissions and reduces environmental performance. According to the previous analysis, corporate digital transformation brings green technological innovation and improved corporate governance, making corporate production and operations greener and more effective. However, with the scale-based expansion of the digital economy, total factor productivity increases carbon emissions by expanding the production scale and energy rebound effect. According to the regression results in column (2) of Table 9, the calculated inflection point value is 119.058, while the average value of the digital transformation index of enterprises in the sample after the expansion of 100 is 0.01, which is far less than the inflection point value, indicating that the degree of digital transformation

of Chinese listed enterprises is not high and has not reached the inflection point. Digital transformation has generally brought positive environmental effects. However, enterprises should also pay more attention to green environmental protection measures during the digital transformation process and should not engage in disorderly development at the expense of the environment.

**Table 9.** Nonlinear Regression of Impact of Corporate Digital Transformation on Environmental Performance.

| Variables | Enp | | | |
|---|---|---|---|---|
| | **(1)** | **(2)** | **(3)** | **(4)** |
| | **SYS-GMM** | | **DIFF-GMM** | |
| $Enp_{t-1}$ | 0.5784 *** | 0.6695 *** | 0.3936 *** | 0.4414 *** |
| | (0.0325) | (0.0009) | (0.0205) | (0.0047) |
| Dig | −0.0121 *** | −0.0086 *** | −0.0093 *** | −0.0021 *** |
| | (0.0008) | (0.0001) | (0.0008) | (0.0002) |
| $Dig^2$ | 1.9290 *** | 2.0478 *** | 1.8914 *** | 0.4690 *** |
| | (0.1463) | (0.0152) | (0.1658) | (0.0380) |
| _cons | 0.0007 *** | −0.0024 *** | 0.0010 *** | 0.0075 *** |
| | (0.0001) | (0.0001) | (0.0001) | (0.0003) |
| Controls | NO | YES | NO | YES |
| AR (1) | 0.0023 | 0.0007 | 0.0117 | 0.0060 |
| AR (2) | 0.2522 | 0.3858 | 0.1666 | 0.3838 |
| Sargan-test | 0.2891 | 0.8993 | 0.2491 | 0.6391 |
| N | 1673 | 1673 | 1307 | 1307 |

Notes: Standard errors in parentheses, *** $p < 0.01$, controls correspond to control variables, AR (1), AR (2), and the Sargan test corresponds to the $p$ value.

## 6. Conclusions and Policy Recommendations

Global environmental governance has always been a critical issue of concern. As the main agents of environmental pollution discharge and environmental governance, enterprises have attached great importance to their social responsibility and environmental protection measures. With the rapid scientific and technological progress and accelerated development of network digitalization, the digital economy, with its high permeability, scale effect and network effect, has become a direct response to the tremendous changes in the internal endowment and external environment of the economy in the new development pattern. This paper takes Chinese A-share listed enterprises as the research object and studies the environmental effects of corporate digital transformation from the perspective of environmental performance. Through the construction of a two-step system GMM measurement model, this paper conducts empirical research, analyzes the environmental effects of corporate digital transformations using the nature of enterprise ownership, enterprise scale, enterprise pollution and regional heterogeneity and analyzes the internal logical relationship between digital transformation and environmental performance from the two channels of green technology innovation and corporate governance. Additionally, the possible nonlinear relationship between corporate digital transformation and environmental performance is also studied. The main conclusions are as follows.

First, corporate digital transformation has effectively curbed environmental pollution emissions and improved environmental performance. After a series of robustness tests, this conclusion is still valid.

Second, from the perspective of enterprises, the environmental effects brought by the digital transformation of state-owned enterprises, large enterprises and heavily polluting enterprises are better than those related to other enterprises. From the perspective of regional heterogeneity, the environmental effects brought by the digital transformation of enterprises in the eastern region are better than in other regions. The impact of digital transformation on environmental performance shows significant heterogeneity.

Third, there is a significant nonlinear relationship between corporate digital transformation and environmental performance. Before the inflection point, digital transforma-

tion can effectively curb environmental pollution emissions and improve environmental performance. When the inflection point is exceeded, digital transformation instead aggravates environmental pollution emissions and reduces environmental performance. The degree of the digital transformation of Chinese enterprises is not high and has not yet reached the inflection point; corporate digital transformation brings more positive environmental effects.

Based on the above research conclusions, this paper puts forward the following policy recommendations. First, enterprises should accelerate their digital transformation to achieve professional production. Through information sharing and big data applications, enterprises should enhance their ability to obtain resources and optimize and upgrade their business models to improve the level of green technology innovation and reduce pollution energy consumption and pollution emissions. Through digital transformation, enterprises should increase the use of renewable energy and improve energy utilization efficiency and green output. Second, enterprises should promote the optimization and upgrading of the organizational structure through digital transformation, improve CSR, strengthen corporate supervision and management, focus on mitigating or resolving the differences in the interests of stakeholders and managers, reduce the cost of principal agent issues, improve their operating ability, achieve more efficient corporate governance, expand the scale of green output, and improve their environmental performance. Third, during the enterprise digital transformation process, more attention should be given to the environmental supervision and management of state-owned enterprises, large enterprises and heavily polluting enterprises. Their digital transformation can bring higher environmental performance, and environmental protection departments can encourage and support these enterprises in further accelerating their digital transformations. In addition, since the economic development and technological level of eastern China are higher than those of the middle and western regions, policy makers can prioritize guiding and supporting corporate digital transformation in the eastern region to improve environmental performance. Through the practical experience of improving environmental performance through digital transformations in the eastern region, the middle and western regions can be driven to achieve digital transformations to improve environmental performance. Fourth, corporate digital transformation does not always bring positive environmental effects. During the digital transformation process, policy makers should actively guide green digital transformation, reduce infrastructure and energy emissions pollution brought by digital transformation, and achieve green development to improve environmental performance and achieve the sustainable development of enterprises and the environment.

The research in this paper enriches the theoretical basis and empirical experience of corporate digital transformation and environmental performance. Of course, there is still room for further improvement. First, because the disclosure of information and the environmental information related to the digital transformation of Chinese listed companies are not perfect, the sample research data are not rich enough, and follow-up research can further optimize the measurement of and fully reflect corporate digital transformation and environmental performance. Second, the transmission mechanism of this paper is based on green technology innovation and corporate governance at the enterprise level. Future research can be conducted from other perspectives, including urban and provincial levels, and can be extended to the environmental effects of digital transformation in emerging or developed markets. Third, this research is mainly based on systematic GMM measurement methods. Future research can use more empirical research tools or policy evaluations to provide a more empirical experience for sustainable environmental development.

**Author Contributions:** Conceptualization, P.X. and L.C.; methodology, P.X.; Software, P.X.; validation, P.X., L.C. and H.D.; formal analysis, P.X. and H.D.; investigation, P.X. and L.C.; resources, P.X. and H.D.; data curation, P.X.; writing—original draft preparation, P.X.; writing—Review and editing, P.X. and H.D.; visualization, P.X.; supervision, P.X. and H.D.; project administration, P.X., L.C. and H.D. All authors have read and agreed to the published version of the manuscript.

**Funding:** This research received no external funding.

**Institutional Review Board Statement:** Not applicable.

**Informed Consent Statement:** Not applicable.

**Data Availability Statement:** The datasets generated and/or analyzed during the current study are available from the corresponding author upon reasonable request.

**Conflicts of Interest:** The authors declare no conflict of interest.

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
