# Peer review of "Pathways to Sustainable Development: Corporate Digital Transformation and Environmental Performance in China"

_sustainability, doi:10.3390/su15010256_

Round 1
Reviewer 1 Report
This interesting paper deals with the digital economy’s rapid development as a new driver of corporate environmental governance and environmental performance improvements, marking a new sustainable development path.
The paper is interesting, well written and organized. Nevertheless, there are still several issues to be solved before the paper can be considered ready for publication.
An important theme that the authors should deepen in the introduction and in the literature review is how service innovation is driven by digital transformation and how this service innovation impacts on environmental performance in the introduction. Service innovation oriented to sustainability should be enhanced by tools supporting managers and entrepreneurs in acknowledging which component of the business model of their firms to unleash sustainability-oriented service innovation. I suggest the authors to consider the paper:
Calabrese, A., Forte, G., & Ghiron, N. L. (2018). Fostering sustainability-oriented service innovation (SOSI) through business model renewal: The SOSI tool. Journal of Cleaner Production, 201, 783-791.
Preghenella, N., & Battistella, C. (2021). Exploring business models for sustainability: A bibliographic investigation of the literature and future research directions. Business Strategy and the Environment, 30(5), 2505-2522.
Moreover, service innovation oriented toward sustainability requires an increasing stakeholder engagement, which should be managed using the appropriate techniques of service process design and representation. See for example:
Proudlove, N. C., Bisogno, S., Onggo, B. S., Calabrese, A., & Ghiron, N. L. (2017). Towards fully-facilitated discrete event simulation modelling: Addressing the model coding stage. European Journal of Operational Research, 263(2), 583-595.
Calabrese, A., & Corbò, M. (2015). Design and blueprinting for total quality management implementation in service organisations. Total Quality Management & Business Excellence, 26(7-8), 719-732.
Finally, the authors address the issue of corporate digital transformation (lines 230-233), please consider the following references:
Annarelli, A., & Palombi, G. (2021). Digitalization Capabilities for Sustainable Cyber Resilience: A Conceptual Framework. Sustainability, 13(23), 13065.
Parida, V., Sjödin, D., & Reim, W. (2019). Reviewing literature on digitalization, business model innovation, and sustainable industry: Past achievements and future promises. Sustainability, 11(2), 391.
Reviewer 2 Report
Dear Authors,
the article you have presented is an interesting discussion of the environmental effects of implementing digital transformation in China. You used a two-stage econometric model.
The paper is very well written, with a rich and substantive introduction, you have thoroughly described the model, the variables, presented the results well, made an attempt to evaluate your work.
In my opinion, the work is suitable for publication. I suggest the authors to add min. 5 scientific publications from the last 10 years, in which the authors use other models in discussing similar issues, supporting environmental mechanisms, agriculture, etc.
Please create a diagram showing the creation of the model and bring it to functionality.
Reviewer 3 Report
The article entitled "Pathways to Sustainable Development: Corporate Digital Transformation and Environmental Performance in China" is of good quality and, in my opinion, does not require any major revisions. But I would appreciate it if authors could add a separate paragraph summarizing the research gaps in previous literature, i.e., what was not researched in the previous works? It should be explicitly shown. The authors should add the paragraph before the paragraph on the current research contributions, Lines 96-112, "The marginal contributions of this paper..."
Good luck!
